# OpenReview forum: "Generative Verifiers: Reward Modeling as Next-Token Prediction"
_NeurIPS.cc/2024/Workshop/MATH-AI — MATH-AI 24_

### Official Review · Reviewer_8MQ6 · 2024-10-03

**Rating:** 9
**Confidence:** 4

**Review:**

## Paper summary

This submission proposes to improve the reasoning capabilities of large language models (LLMs) by training verifiers (reward models) as generative models rather than as standard discriminative models. Instead of simply scoring solutions, these "Generative Verifiers" (GenRM) leverage next-token prediction to verify solutions through language generation. This approach leads to better performance on tasks like math reasoning and algorithmic problems compared to baselines.

## Strengths
- The paper reads well and is easy to follow. I enjoyed reading the paper.
- The paper clearly illustrates the benefits of GenRM via experiments.

## Areas for improvement
- In the abstract, the authors claim GenRM integrates seamlessly with instruction tuning, but in the main paper there are not enough relevant discussions. Does this refer to Selection 2.2 (joint training on $\mathcal D_{\text{verify}}$ and $\mathcal D_{\text{correct}}$)?
- This idea of utilizing additional inference-time compute with GenRM is very interesting. Some recent papers [1,2,3,4] discussed similar ideas. While I understand that those works are concurrent, it would be nice if the authors can add some discussions and comparisons, e.g., on why scaling inference-time compute of the reward model can be benifitial.
- Open-source: Do the authors plan to release relevant code and CoT data to the community?

[1] Large Language Monkeys: Scaling Inference Compute with Repeated Sampling

[2] An Empirical Analysis of Compute-Optimal Inference for Problem-Solving with Language Models

[3] Scaling LLM Test-Time Compute Optimally can be More Effective than Scaling Model Parameters

[4] Smaller, Weaker, Yet Better: Training LLM Reasoners via Compute-Optimal Sampling

---

### Official Review · Reviewer_yFNJ · 2024-10-07
**Generative Verifiers: Reward Modeling as Next-Token Prediction**

**Rating:** 8
**Confidence:** 4

**Review:**

Pros:
Innovative Approach: The paper introduces a novel method of training verifiers using a next-token prediction model, which is a significant shift from the traditional discriminative classifiers approach. This leverages the inherent text generation capabilities of large language models (LLMs).
Enhanced Performance: The proposed Generative Reward Modeling (GenRM) demonstrates superior performance over traditional methods, with improvements ranging from 16% to 64% in problem-solving tasks. This is a substantial enhancement, showcasing the practical effectiveness of the approach.
Integration with LLMs: GenRM integrates seamlessly with existing LLM architectures, benefiting from chain-of-thought reasoning and majority voting mechanisms. This integration not only simplifies the implementation but also enhances the reasoning capabilities of the models.
Scalability: The paper provides evidence that GenRM scales favorably with increases in dataset size, model capacity, and inference-time compute, making it a robust solution for various applications.
Detailed Experimental Analysis: The experiments are well-designed and comprehensive, covering multiple reasoning tasks and comparing the new method with several baseline approaches. The use of figures and tables to illustrate the performance improvements is particularly helpful.
Cons:
Complexity in Implementation: While the paper outlines the benefits and the theoretical framework of GenRM, the actual implementation and tuning might be complex due to the integration of multiple advanced techniques like chain-of-thought reasoning and majority voting.
Potential Overfitting: The method's reliance on next-token prediction and the integration of multiple functionalities within a single model could lead to overfitting, especially on smaller datasets or more niche tasks.

Conclusion:
The paper presents a compelling new approach to reward modeling in LLMs, showing significant potential to enhance the reasoning capabilities of these models. While there are challenges related to implementation complexity and computational demands, the innovative use of next-token prediction for verification tasks marks a notable advancement in the field. Future work could focus on addressing these challenges and expanding the applicability of GenRM to a wider array of tasks.

---

### Official Review · Reviewer_XS6T · 2024-10-08
**This work is of high quality and introduces a novel approach to enhancing the reward model. It suggests that instructing the model to provide chain-of-thought (CoT) reasoning before predicting correctness can outperform direct label prediction.**

**Rating:** 7
**Confidence:** 4

**Review:**

This work is of high quality and introduces a novel approach to enhancing the reward model. It suggests that instructing the model to provide chain-of-thought (CoT) reasoning before predicting correctness can outperform direct label prediction.

---

### Official Review · Reviewer_qXxN · 2024-10-08
**Not as novel as claimed but a well-executed analysis of reward modeling choices**

**Rating:** 6
**Confidence:** 4

**Review:**

## Summary
The paper suggests to train reward models with a cross-entropy loss on two tokens found natively in the model's vocabulary. In this way, the reward modeling objective can naturally be combined with Chain-of-Thought (CoT) inference. When trained jointly on reward modeling (critic) and generation (policy) samples distilled from a larger model, a performant verifier is obtained that scales favorably with model size, training dataset  size and inference budget.

## Weaknesses:
- Using a reward that is modeled with two tokens as $\mathrm{sigmoid}(x(1) - x(0))$ instead of with one token as $\mathrm{sigmoid(x(1))}$ is by no means original. It appears, for instance, in Lample et al. 2022 Section 5.1, and is considered "folk" knowledge by practitioners. (The reasoning being that transformers prefer directions to magnitudes due to their normalization layers, not that pretrained language modeling priors play a large role.)
- Coupling reward models with CoT inference is a useful idea, but has been explored extensively in prior works (the cited related work Ye et al. 2024 or Li et al. 2023 for instance).
- The main contribution of GenRMs appears to be the joint training of policy and critic (which is standard in RL, cf. Silver et al. 2017; Lample et al. 2022) and the distillation of solution CoTs from a larger model that rationalizes ground-truth solutions to a point that the trained "reward model" outperforms the policy model it is supposed to supervise (Fig.D1 80% GSM8K vs 73% for Gemini 1.0 Pro in introduction).
- paper writing: the paper is likely the introduction and method section of a full conference submission and moved ALL experiments and results to appendices. The paper should be rewritten to summarize main findings in the main text body. The introduction and method section are disproportionately long, it could have been explained in few lines and one equation.

## References
Lample, Guillaume, et al. "Hypertree proof search for neural theorem proving." Advances in neural information processing systems 35 (2022): 26337-26349. https://arxiv.org/abs/2205.11491

Ye, Zihuiwen, et al. "Improving Reward Models with Synthetic Critiques." arXiv preprint arXiv:2405.20850 (2024). https://arxiv.org/abs/2405.20850

Li, Junlong, et al. "Generative judge for evaluating alignment." arXiv preprint arXiv:2310.05470 (2023). https://arxiv.org/abs/2310.05470

Silver, David, et al. "Mastering chess and shogi by self-play with a general reinforcement learning algorithm." arXiv preprint arXiv:1712.01815 (2017). https://arxiv.org/abs/1712.01815

---

### Official Review · Reviewer_bJoL · 2024-10-09
**The paper proposes a new reward modeling but additional experiments are needed to better assess its effectiveness.**

**Rating:** 6
**Confidence:** 3

**Review:**

Pros:
+ The paper introduces a new perspective on training verifiers by recasting reward modeling as next-token prediction.
+ The experimental setup is fair: Experiments are conducted on multiple tasks to validate the approach and the method is compared against several common baselines.
+ The paper also details model architectures, hyperparameters, and training procedures, which aids in reproducibility.
+ The paper is well-written and easy to follow.

Cons:
- When evaluating GenRM-COT's data scaling on GSM8K, the number of rationales per problem is limited.
- The paper doesn’t compare GenRM with some recently advanced verification works like Lightman et al 2023, Let’s verify step by step.
- To better evaluate GenRM-COT's effectiveness and consistency, additional experiments on output variance would be beneficial.

---

### Decision · Program_Chairs · 2024-10-08

**Decision:**

Accept

**Comment:**

All reviewers are in support of this work's acceptance, and it will make a great contribution to our workshop.